# Geographic variation in COVID-19 vulnerability by legal immigration status in California: a prepandemic cross-sectional study

Heeju Sohn [1], Jasmine Ko Aqua [2]

¹Sociology, Emory University, Atlanta, Georgia, USA
²Department of Epidemiology, Rollins School of Public Health, Emory University, Atlanta, Georgia, USA

**Correspondence to**
Dr Heeju Sohn;
heeju.sohn@emory.edu

## ABSTRACT

**Objective** To quantify COVID-19 vulnerabilities for Californian residents by their legal immigration status and place of residence.

**Design** Secondary data analysis of cross-sectional population-representative survey data.

**Data** All adult respondents in the restricted version of the California Health Interview Survey (2015–2020, n=128 528).

**Outcome measure** Relative Social Vulnerability Indices for COVID-19 by legal immigration status and census region across six domains: socioeconomic vulnerability; demography and disability; minority status and language barriers; high housing density; epidemiological risk; and access to care.

**Results** Undocumented immigrants living in Southern California's urban areas (Los Angeles, Orange, San Diego-Imperial) have exceptionally high vulnerabilities due to low socioeconomic status, high language barriers, high housing density and low access to care. San Joaquin Valley is home to vulnerable immigrant groups and a US-born population with the highest demographic and epidemiological risk for severe COVID-19.

**Conclusion** Interventions to mitigate public health crises must explicitly consider immigrants' dual disadvantage from social vulnerability and exclusionary state and federal safety-net policies.

## INTRODUCTION

The novel coronavirus, SARS-CoV-2, has spread across all parts of the USA, exacerbating entrenched social and health inequalities in its wake. This article uses sensitive immigration and geographic information from the restricted data in the California Health Interview Survey (CHIS) to quantify underlying socioeconomic, demographic and epidemiological vulnerabilities to COVID-19 by legal immigration status in California's 10 census regions.

Prior research on immigrants' resources and health suggests that their vulnerability to COVID-19 may be higher than the US-born population.[1] Higher prevalence of health conditions such as obesity, asthma

## STRENGTHS AND LIMITATIONS OF THIS STUDY

⇒ We adapted the Centers for Disease Control and Prevention Social Vulnerability Index to quantify immigrants' vulnerability to COVID-19 by their legal immigration status and their geographic region of residence in California.

⇒ Our analysis used the California Health Interview Survey (2015–2020, n=128 528), which contains direct measurements of immigrants' legal status as well as detailed socioeconomic and health information.

⇒ The data cover 2015–2020, and vulnerability indices may diverge from the pandemic's peaks in 2021.

⇒ Vulnerability indices are relative measures among California's 50 immigrant status-region groups and cannot be generalised to the broader national population.

and diabetes among socioeconomically disadvantaged groups[2 3] suggests that immigrants may also have a higher risk for severe COVID-19 outcomes.[4] Many immigrants work in occupations that required in-person work throughout the pandemic[5 6] and live in larger households making isolation difficult,[7] which indeed became an issue as the pandemic progressed, as commentaries describe health officials frequently discovering up to 10 workers sharing a two-bedroom apartment or several families living in one house, most sick by the time contact tracers were able to notify them.[8] At the same time, immigrants comprise a large and diverse group,[9 10] in which some subgroups have high levels of education and income, whereas other subgroups have high rates of poverty and economic insecurity. Unequal distribution of healthcare resources across geographic regions and residential segregation may also contribute to inequities in COVID-19 mortality within immigrant communities.[11] Yet, systematic and precise information on immigrants' vulnerabilities is absent from policy making due to the

lack of detailed immigration information in population-representative health surveys.

As the COVID-19 pandemic progressed, commentaries and limited research described the unique difficulties and high risk for adverse COVID-19 outcomes that undocumented immigrants faced. Reports from hospital emergency rooms detailed inpatient teams struggling to communicate with Spanish-speaking patients using language lines and through layers of personal protective equipment.[8] In addition to reports of language barriers, a cross-sectional survey of adult, Spanish-speaking, non-citizen Latinx immigrants found that a substantial percentage of participants would not identify an undocumented household member or coworker during contact tracing, believed that uninsured immigrants were limited to hospital emergency departments for COVID-19 testing or treatment and agreed that using public COVID-19 testing and treatment services could jeopardise an individual's immigration prospects.[12] Reports also noted high COVID-19 case rates and numerous significant outbreaks in federal immigrant detention centres,[13–15] as well as a fear that convention centres that served as COVID-19 treatment facilities were actually immigration detention centres.[8]

In addition to the COVID-19 risk factors and other unique difficulties described above, undocumented immigrants also face greater structural barriers in accessing healthcare and safety-net programmes.[16] Federal policies dictate immigrants' access to federally funded healthcare services and safety-net programmes[17 18] based on their immigration status. The March 2020 Coronavirus Aid, Relief, and Economic Security Act explicitly barred undocumented immigrants from receiving direct federal financial relief, continuing the long-standing policy of barring undocumented immigrants from cash assistance.[19] Legally present visa holders and immigrants who have permission to live and work in the USA also have limited access to resources they can turn to during crises. The 1996 Personal Responsibility and Work Opportunity Reconciliation Act restricted legal immigrants' eligibility for federally funded safety-net programmes,[17] and US Citizenship and Immigration Services (USCIS) issued guidance at the beginning of the pandemic that legal immigrants could be denied citizenship or permanent residency for receiving an expanded range of eligible public healthcare benefits.[20 21] Though USCIS issued an alert on 14 March 2020, that COVID-19 testing, treatment, preventive care and vaccines (when available) would be exempt from the 'public charge' admissibility assessment and stopped applying this Public Charge Final Rule on 9 March 2021,[22] communication and implementation of these changes remained unclear, and many immigrants still believed that using public COVID-19 testing and treatment services could jeopardise their immigration prospects.[12]

When COVID-19 vaccines became available to the general public, the Department of Homeland Security released a statement that they fully support equal access to the vaccine for undocumented immigrants.[23] Despite this public statement, the majority of state public health websites do not explicitly mention this information and many reports emerged throughout 2021 of undocumented immigrants being asked to provide Social Security numbers at vaccination sites or being turned away for not presenting a state-issued ID.[24–26] Many undocumented immigrants also occupy jobs without paid leave and have language barriers that can impact their comprehension of vaccine information and education, introducing additional structural barriers to vaccination.

Policies at the state and local levels can support inclusive public health programmes and outreach to address their immigrant communities' specific needs.[27 28] Some localities have expanded healthcare services to undocumented immigrant children and low-income pregnant women, allowed for in-state tuition and financial aid for undocumented students and issued government identification to all residents.[29] Inclusive policies facilitate schooling and employment[30] for vulnerable immigrant groups and have been linked to better health outcomes.[31] California became the first state to provide COVID-19 disaster relief assistance to undocumented adults who are ineligible for other forms of assistance, providing a one-time direct assistance payment of $500 (maximum of $1000 per household).[32] This $75 million disaster relief assistance programme was estimated to reach 150 000 undocumented Californians through 12 immigrant-serving non-profit organisations.[32] The demand for relief quickly overwhelmed the available resources, with reports of people unable to get through phone lines due to extremely high call volumes and equating the direct assistance to 'winning the lottery'.[33]

Conversely, exclusionary policies such as mandating the use of E-Verify, an electronic database of immigrants' work authorisation, or barring states from issuing drivers' licences, or granting college admission to undocumented immigrants aim to create obstacles for those who do not have legal status.[29 34] Localities that coordinate with immigration enforcement also deter many immigrants and their families from seeking help regardless of their citizenship status.[35 36]

This article identifies opportunities for local-level and community-level interventions that can address the immigrants' unique challenges. Fractured policies that stratify people by immigration status stymie efforts that aim to mitigate the effects of the COVID-19 pandemic for all.

## METHODS

### Data source

We used the 2015–2020 survey data from CHIS, a collaborative data collection between UCLA Center for Health Policy Research, the California Department of Public Health and the Department of Health Care Service.[37] The CHIS is a large, annual random-digit telephone survey of public health and healthcare access issues in California and is one of few representative surveys of this scale that

collected information on detailed immigrant documentation status uncommon in large-scale surveys. The survey aims to produce estimates for under-represented immigrant subgroups and administers the questionnaire in Spanish, Vietnamese, Korean, Mandarin, Cantonese and Tagalog in addition to English.

### Study population

Our analysis included 128 528 adult survey respondents 18 years or older and used individual weights to account for sampling design. The CHIS imputed missing values for almost all variables in their surveys using random selection or hot deck imputation used in census-published data sets.

### Patient and public involvement

The article presents analyses of secondary survey data, and no patients were involved in the study.

### Documentation status

We categorised respondents by their nativity and legal immigration status: US-born citizens, naturalised citizens, legal permanent residents (LPR), documented temporary visa holders and undocumented immigrants. The first three categories, which accounted for almost 97% of our analysis sample, were determined directly for the entire study period 2015–2020 through a series of citizenship and immigration questions. All respondents answered whether they were born in the USA. If the response was no, they indicated whether they were naturalised citizens. Respondents who were not US citizens were then asked whether they were LPRs. Questions that can differentiate undocumented immigrants from documented temporary visa holders (non-LPRs) were only asked in 2015–2016. The large majority (98.4%) of our analysis sample had direct information on immigration status, including whether they were undocumented or living in the USA on valid visas. The remaining 1.6% of our sample non-citizens who were not LPRs in the years 2017–2020 accounted for about half of non-citizens and non-LPRs. They may have had a valid visa to live in the USA (ie, students and diplomats), but CHIS did not ask for specific details on visa status during those survey years. We used a multiple imputation procedure to differentiate the documented temporary visa holders from the likely undocumented based on the relationships between sociodemographic characteristics and documentation status derived from the complete survey years in 2015 and 2016.[38 39] We included age, age-squared, sex, educational attainment, country of origin, family type, English proficiency, years lived in the USA, federal poverty level and geographic location in our multiple imputation procedure.[40] The imputation method to differentiate undocumented immigrants from documented temporary visa holders has been applied in national surveys such as the Survey of Income and Program Participation.[41] It has also been applied to impute immigration status in a 'recipient' survey (American Community Survey) using

data from a 'donor' survey that directly collected immigration information.[42 43] These approaches are an extension of multiple imputation methods that leverage the relationships between variables with missing and known characteristics.[44]

### COVID-19 vulnerability index

We adapted the validated US Centers for Disease Control and Prevention (CDC) Social Vulnerability Index (SVI)[45] to develop a COVID-19 vulnerability index. We modified the CDC SVI based on the variables available in our data set and expanded the index to include additional factors critical to the COVID-19 pandemic.[46] The first four themes in our COVID-19 vulnerability index (socioeconomic, demographic/disability, minority and language, and housing density) are based on CDC SVI. We were not able to include two factors from CDC SVI—physical/mental/emotional disability status and vehicle ownership—as CHIS did not ask these questions during the study period. Instead, we included a factor that indicated serious psychological distress based on Kessler's Psychological Distress Questionnaire.[47] We also added a factor variable indicating the proportion of respondents living in an urban area to augment the CDC's housing density theme. In addition to the CDC SVI's four original themes, our analysis uses CHIS' detailed health questionnaire and examined two more themes: epidemiological factors and access to healthcare. Overall, we incorporated 21 factors across six domains in our COVID-19 vulnerability index. Table 1 lists the six themes and their factors.

### Domain 1: socioeconomic vulnerability

This domain captures the disproportionate crisis vulnerability associated with economic disadvantage. Households living below the poverty line face increased COVID-19 vulnerability due to structural health inequities and disproportionate distribution of underlying comorbidities.[48 49] Individuals with higher educational attainment have greater access to and may better adapt to COVID-19 risk communications and health messaging.[50]

### Domain 2: demographic vulnerability and disability

This domain captures the increased danger that vulnerable demographic groups face in disaster situations such as the COVID-19 pandemic. Older adults are at greater risk of requiring hospitalisation or dying if diagnosed with COVID-19, and single parents and individuals with disabilities may experience additional stressors of the pandemic.[51] The pandemic has been particularly challenging for single-parent households, where only one parent is available for multiple responsibilities that may include working extra shifts, caring for a sick family member or supervising online schooling.

### Domain 3: minority status and language barriers

This domain captures minority and marginalised populations' disproportionate vulnerability. About 33% of US-born and 60% of the foreign-born population in California self-reported as non-white,[52] and they may

**Table 1** COVID-19 Social Vulnerability Index (SVI) domains and factors

| | Domain | Factors | Description |
|---|---|---|---|
| 1 | Socioeconomic vulnerability* | Below poverty level* | Calculated as the proportion of households at 0%–99% federal poverty level |
| | | Unemployed* | Calculated as the proportion of households with both respondent and spouse (if present) unemployed |
| | | No high school diploma* | Calculated as the proportion of respondents with less than a high school diploma |
| 2 | Demographic vulnerability and disability* | Aged 65 or older* | Calculated as the proportion of respondents aged 65 or older |
| | | Single-parent household* | Calculated as the proportion of single-parent households with children under 18 years old |
| | | Psychological disability† | Calculated as the proportion of respondents with a score of 13 or above on the Kessler Psychological Distress Scale |
| 3 | Minority status and language barriers* | Minority* | Calculated as the proportion of non-white race or Hispanic ethnicity respondents |
| | | Non-English speaker* | Calculated as the proportion of respondents who speak English 'not well' or 'not at all' |
| 4 | High housing density* | Multiunit structures/mobile homes* | Calculated as the proportion of respondents who live in a multifamily or mobile house |
| | | Urbanisation‡ | Calculated as the proportion of respondents who live in an urban or metropolitan area |
| | | Extended household† | Calculated as the proportion of households with three or more adults |
| 5 | Epidemiological risk factors§ | High blood pressure§ | Calculated as the proportion of respondents with ever physician-diagnosed high blood pressure |
| | | Heart disease§ | Calculated as the proportion of respondents with ever physician-diagnosed heart disease |
| | | Asthma§ | Calculated as the proportion of respondents who reported currently having asthma |
| | | Smoking§ | Calculated as the proportion of respondents who reported being a current or former smoker |
| | | Obesity§ | Calculated as the proportion of respondents with a BMI of 30 or more for non-Asians or 27 or more for Asians |
| | | Diabetes§ | Calculated as the proportion of respondents with ever physician-diagnosed diabetes |
| | | Healthcare occupation† | Calculated as the proportion of respondents with an occupation in healthcare delivery |
| | | High-risk occupation† | Calculated as the proportion of respondents in essential occupations that have high risk of exposure to infectious diseases |
| 6 | Low access to healthcare‡ | No health insurance† | Calculated as the proportion of respondents who reported having no health insurance in the past 12 months |
| | | No usual source of healthcare† | Calculated as the proportion of respondents who reported no usual source of healthcare (ie, doctor's office, community or government clinic, community hospital) |

*Adapted from Centers for Disease Control and Prevention (CDC) Social Vulnerability Index (SVI).
†Author included.
‡Adapted from Acharya and Porwal.[45]
§Adapted from Surgo Foundation's COVID-19 Community Vulnerability Index (CCVI).
BMI, body mass index.

encounter more racialised discrimination in healthcare settings than their white counterparts.[53] Limited English proficiency can be a barrier to accessing health services and understanding COVID-19 health messaging; recent studies linked low English proficiency with an increased risk of COVID-19.[49]

### Domain 4: high housing density
We included density factors associated with an increased risk of SARS-CoV-2 transmission: the proportion of respondents who live in a multifamily or mobile house, the proportion of respondents who live in an urban or metropolitan area and the proportion of households with three or more adults.[54]

### Domain 5: epidemiological risk factors
This domain captures the medical and epidemiological risk factors associated with COVID-19 infection and its adverse outcomes. The medical risk factors for severe COVID-19 in this domain include cardiovascular conditions (high blood pressure and heart disease), respiratory conditions (asthma and smoking), obesity and diabetes.[51] Epidemiological risk factors included occupations with a high risk of COVID-19 exposure. We used the O*NET's Work Surveys to identify high-risk occupations and cross-referenced them with California's Executive Order N-33-20 that defined essential workers. We harmonised the occupation categories with CHIS and assigned occupations in healthcare, service, transportation, construction and extraction in the high-risk category.

### Domain 6: low access to healthcare
This domain encapsulates the additional vulnerability that healthcare barriers, such as the lack of health insurance, add during a widespread health crisis. Concerns about the cost of testing and treatment and uncertainty around where to seek medical attention lead to delayed patient care and disrupt our ability to control epidemics.[50]

We constructed a vulnerability index for each of the six domains by immigration status intersected with census region (5 immigrant groups × 10 regions=50 immigrant-region groups). First, we estimated groups' proportions in the high vulnerability category for each of the 21 factors. Second, we averaged the proportions across factors within each domain. Third, we ranked immigrant status-region groups from the group with the highest proportion in the vulnerable category to the lowest. We then assigned a percentile rank using the following equation: Percentile Rank = (rank–1)/(N-1), where N equals 50 and represents the total number of immigrant status-region groups. A higher percentile indicates greater relative vulnerability. Our approach is the same as the method used by Acharya and Porwal,[45] and Flanagan and colleagues.[54]

## RESULTS

Table 2 summarises the demographic and socioeconomic characteristics by immigration status across California. The values are weighted by population and largely reflect the profiles of people living in urban areas. Similar to previous state-wide studies, documented temporary visa holders tended to be younger and healthier than other immigrant groups. Naturalised citizens are older than other groups with more health conditions than other immigrants. At the same time, they are less likely to live in poverty or without health insurance.

Table 3 reports the vulnerability indices in six domains for five immigrant groups living in California's 10 census regions. Indices range from 0 (least vulnerable) to 1 (most vulnerable) and represent the relative vulnerability within 50 immigrant-region groups.

Undocumented immigrants have high vulnerability due to low socioeconomic status, the concentration of minorities and language barriers and low access to care across the entire state. Undocumented immigrants living in the San Joaquin Valleys have the highest socioeconomic vulnerability. In contrast, vulnerability due to minority status and language barriers is the highest among undocumented immigrants in San Diego County (0.98) and Central Coast (1.00).

Naturalised citizens and US-born citizens share similar vulnerability profiles across the 10 regions, but unlike non-citizen immigrants, their sources of vulnerability are predominantly from demographic composition and disability. They also score high in vulnerability from epidemiological COVID-19 risk factors, especially in the North Coast and the San Joaquin Valleys.

The wide range of vulnerability indices across California's regions reflects documented temporary visa holders' socioeconomic and demographic diversity that was obscured in table 2. Documented temporary visa holders living in the San Francisco Bay Area are among the least vulnerable—they are the most socioeconomically and demographically advantaged (indices of 0.0) with low epidemiological risk for COVID-19. Conversely, documented temporary visa holders living in Southern San Joaquin Valley have a high socioeconomic vulnerability and low access to healthcare.

Vulnerability due to high housing density is concentrated among non-citizen immigrants, including LPRs in Southern California—Los Angeles County, Orange County and San Diego-Imperial—and is likely linked to high housing costs in these regions. San Joaquin Valley is home to vulnerable non-citizen immigrants, including LPRs, due to their low socioeconomic status, high minority populations and language barriers.

Table 3 also reports the overall vulnerability that combines all six domains, and the last column in the table indicates its ranking among the 50 immigrant status-region groups. Undocumented immigrants living in Southern California (Los Angeles County, Orange County and San Diego-Imperial regions) had the highest overall vulnerability. US-born citizens and documented temporary visa holders in regions near San Francisco—San Francisco Bay Area, North Coast and Central Coast—scored the lowest in overall vulnerability.

Table 4 presents the *concentration of vulnerability* for each immigrant status-region group. The values in table 4 indicate the number of vulnerability themes out of a possible six that scored in the top 75th percentile. Table 5 presents a full correlation table between the six themes with tests of statistical significance. Groups whose vulnerability stems from low socioeconomic status are likely to share vulnerabilities from being a member of a minority group, experiencing language barriers (R=0.858) and having low access to healthcare (R=0.561). Groups' minority populations and language barriers are also correlated with high housing density (R=0.574) and low access to healthcare (R=0.757). High epidemiological and demographic vulnerabilities were not significantly correlated with high vulnerabilities from social causes.

Naturalised citizens had the fewest high-scoring (top 75th percentile) vulnerabilities with an average of 0.7 across 10 regions. Undocumented immigrants had the most high-scoring vulnerabilities. Undocumented immigrants living in urban centres surrounding San Francisco, Los Angeles and San Diego scored in the top 75th percentile for five out of six vulnerability domains. They also had a high concentration of vulnerabilities (four out of six) in non-urban regions where the vulnerability was low for other groups such as Superior California. North Coast and the Inland Empire regions had a relatively high concentration of vulnerability due to high scores among US-born citizens and naturalised citizens in addition to immigrants with liminal statuses.

## DISCUSSION

Our study highlights the unequal social vulnerabilities between people with different legal immigration statuses across California during the years leading up to the COVID-19 pandemic. Our domain-specific analyses showed that vulnerabilities from low socioeconomic status, language barriers, high housing density and low

**Table 2** Demographic and socioeconomic characteristics of California residents by legal immigration status (2015–2020)

| | | US-born citizens n=100 387 (78.1%) | Naturalised citizens n=18 386 (14.3%) | Legal permanent residents (LPR) n=5825 (4.5%) | Documented temporary visa holders n=2813 (2.2%) | Undocumented immigrants n=1117 (0.9%) |
|---|---|---|---|---|---|---|
| **Demographic characteristics** | | | | | | |
| Mean age | * | 46.2 | 52.5 | 45.6 | 33.9 | 38.9 |
| Mean family size | | 2.1 | 2.3 | 2.6 | 2.3 | 3.0 |
| Mean household size | | 3.1 | 3.5 | 3.8 | 3.1 | 4.5 |
| Mean years lived in the USA | | na | 31.4 | 20.0 | 7.6 | 16.7 |
| Proportion of female | | 0.51 | 0.53 | 0.50 | 0.48 | 0.47 |
| Proportion of non-white or Hispanic | † | 0.45 | 0.85 | 0.90 | 0.86 | 0.99 |
| Proportion in households with 3 or more adults | ‡ | 0.40 | 0.50 | 0.52 | 0.31 | 0.54 |
| Proportion living in single-parent household | * | 0.07 | 0.06 | 0.07 | 0.08 | 0.17 |
| Proportion living in urban area | ‡ | 0.97 | 0.99 | 0.99 | 1.00 | 0.99 |
| **Socioeconomic characteristics** | | | | | | |
| Proportion with household incomes below 100 % Federal Poverty Level (FPL) | § | 0.12 | 0.16 | 0.25 | 0.21 | 0.46 |
| Proportion with no earners in family | § | 0.30 | 0.27 | 0.19 | 0.17 | 0.16 |
| Proportion without a high school (HS) degree or equivalent | § | 0.07 | 0.26 | 0.44 | 0.11 | 0.64 |
| Proportion living in a multiunit structure or a mobile home | ‡ | 0.13 | 0.15 | 0.19 | 0.11 | 0.30 |
| Proportion without health insurance | ¶ | 0.29 | 0.30 | 0.42 | 0.66 | 0.56 |
| Proportion with no usual source of healthcare | ¶ | 0.06 | 0.07 | 0.13 | 0.20 | 0.42 |
| Proportion in healthcare-related occupation | ** | 0.03 | 0.03 | 0.01 | 0.02 | 0.00 |
| Proportion in occupations with close physical contact with others | ** | 0.15 | 0.14 | 0.26 | 0.33 | 0.41 |
| Proportion who speaks English not well or not at all | † | 0.01 | 0.26 | 0.49 | 0.20 | 0.72 |
| **Health characteristics** | | | | | | |
| Proportion with fair or poor self-rated health | | 0.16 | 0.24 | 0.29 | 0.11 | 0.34 |
| Proportion scoring above the threshold for psychological distress in the past 12 months | * | 0.20 | 0.23 | 0.10 | 0.00 | 0.02 |
| Proportion with at least one comorbid condition: asthma, diabetes, cardiovascular disease, high blood pressure, obesity, current/former smoker | ** | 0.61 | 0.56 | 0.56 | 0.42 | 0.59 |

Sample is limited to adults aged 18 and over. Distinction between documented temporary visa holders and undocumented immigrants for years 2017–2020 is derived from multiple imputation using complete data in years 2015 and 2016.
Source: Authors' analysis of the restricted data from the California Health Interview Survey (2015–2020).
*Included in demographic vulnerability and disability domain.
†Included in minority status and language barrier domain.
‡Included in high housing density domain.
§Included in socioeconomic vulnerability domain.
¶Included in healthcare access domain.
**Included in epidemiological risk domain.
na, not applicable.

**Table 3** Domain-specific and overall social and COVID-19 vulnerability indices by California census region and immigrant status group

| Region* | Immigrant status | n (unweighted) | Socioeconomic vulnerability | Demographic vulnerability and disability | Minority status and language barriers | High housing density | Epidemiological risk factors | Low access to healthcare | Overall vulnerability | Rank† |
|---|---|---|---|---|---|---|---|---|---|---|
| Superior California | US-born citizens | 16588 | 0.163 | 0.898 | 0.020 | 0.020 | 0.735 | 0.122 | 0.041 | 48 |
| | Naturalised citizens | 1357 | 0.429 | 0.612 | 0.265 | 0.122 | 0.449 | 0.184 | 0.265 | 37 |
| | Legal permanent residents (LPR) | 442 | 0.653 | 0.245 | 0.612 | 0.224 | 0.184 | 0.490 | 0.469 | 27 |
| | Documented temporary visa holders | 58 | 0.469 | 0.469 | 0.429 | 0.510 | 0.041 | 0.735 | 0.347 | 33 |
| | Undocumented immigrants | 183 | 0.878 | 0.163 | 0.796 | 0.571 | 0.837 | 0.898 | 0.857 | 8 |
| North Coast | US-born citizens | 7455 | 0.122 | 1.000 | 0.000 | 0.000 | 0.857 | 0.143 | 0.000 | 50 |
| | Naturalised citizens | 472 | 0.490 | 0.980 | 0.224 | 0.041 | 0.918 | 0.041 | 0.306 | 35 |
| | LPR | 189 | 0.633 | 0.510 | 0.673 | 0.163 | 0.980 | 0.245 | 0.633 | 19 |
| | Documented temporary visa holders | 26 | 0.837 | 0.061 | 0.204 | 0.327 | 0.000 | 0.000 | 0.143 | 43 |
| | Undocumented immigrants | 97 | 0.796 | 0.122 | 0.878 | 0.449 | 1.000 | 0.776 | 0.878 | 7 |
| San Francisco Bay Area | US-born citizens | 15060 | 0.020 | 0.776 | 0.102 | 0.245 | 0.429 | 0.082 | 0.061 | 47 |
| | Naturalised citizens | 3728 | 0.184 | 0.653 | 0.347 | 0.367 | 0.204 | 0.020 | 0.245 | 38 |
| | LPR | 949 | 0.265 | 0.204 | 0.551 | 0.673 | 0.265 | 0.449 | 0.408 | 30 |
| | Documented temporary visa holders | 400 | 0.000 | 0.000 | 0.245 | 0.694 | 0.020 | 0.510 | 0.020 | 49 |
| | Undocumented immigrants | 336 | 0.755 | 0.551 | 0.898 | 0.959 | 0.898 | 0.796 | 0.898 | 6 |
| Northern San Joaquin Valley | US-born citizens | 6215 | 0.327 | 0.939 | 0.122 | 0.061 | 0.939 | 0.327 | 0.163 | 42 |
| | Naturalised citizens | 628 | 0.592 | 0.694 | 0.408 | 0.082 | 0.531 | 0.102 | 0.388 | 31 |
| | LPR | 341 | 0.776 | 0.265 | 0.755 | 0.306 | 0.551 | 0.571 | 0.714 | 15 |
| | Documented temporary visa holders | 16 | 0.510 | 0.020 | 0.653 | 0.592 | 0.082 | 0.673 | 0.449 | 28 |
| | Undocumented immigrants | 195 | 0.980 | 0.306 | 0.939 | 0.633 | 0.347 | 0.755 | 0.837 | 9 |
| Central Coast | US-born citizens | 7859 | 0.061 | 0.878 | 0.041 | 0.184 | 0.388 | 0.204 | 0.082 | 46 |
| | Naturalised citizens | 997 | 0.388 | 0.673 | 0.327 | 0.469 | 0.367 | 0.224 | 0.367 | 32 |
| | LPR | 404 | 0.694 | 0.531 | 0.735 | 0.755 | 0.571 | 0.612 | 0.735 | 14 |
| | Documented temporary visa holders | 54 | 0.306 | 0.592 | 0.449 | 0.735 | 0.163 | 0.714 | 0.571 | 22 |
| | Undocumented immigrants | 275 | 0.939 | 0.102 | 1.000 | 0.796 | 0.612 | 0.980 | 0.939 | 4 |
| Southern San Joaquin Valley | US-born citizens | 6386 | 0.347 | 0.959 | 0.143 | 0.102 | 0.959 | 0.347 | 0.224 | 39 |
| | Naturalised citizens | 724 | 0.612 | 0.755 | 0.510 | 0.143 | 0.755 | 0.388 | 0.490 | 26 |
| | LPR | 376 | 0.816 | 0.388 | 0.816 | 0.347 | 0.673 | 0.531 | 0.796 | 11 |
| | Documented temporary visa holders | 29 | 0.531 | 0.041 | 0.571 | 0.551 | 0.122 | 0.857 | 0.612 | 20 |
| | Undocumented immigrants | 292 | 1.000 | 0.286 | 0.959 | 0.429 | 0.224 | 0.837 | 0.816 | 10 |

Continued

**Table 3** Continued

| Region* | Immigrant status | n (unweighted) | Socioeconomic vulnerability | Demographic vulnerability and disability | Minority status and language barriers | High housing density | Epidemiological risk factors | Low access to healthcare | Overall vulnerability | Rank† |
|---|---|---|---|---|---|---|---|---|---|---|
| Inland Empire | US-born citizens | 8068 | 0.204 | 0.918 | 0.163 | 0.204 | 0.816 | 0.367 | 0.204 | 40 |
| | Naturalised citizens | 1313 | 0.571 | 0.796 | 0.531 | 0.286 | 0.796 | 0.429 | 0.592 | 21 |
| | LPR | 463 | 0.735 | 0.327 | 0.776 | 0.388 | 0.694 | 0.633 | 0.755 | 13 |
| | Documented temporary visa holders | 55 | 0.286 | 0.184 | 0.592 | 0.653 | 0.592 | 0.816 | 0.673 | 17 |
| | Undocumented immigrants | 210 | 0.959 | 0.490 | 0.837 | 0.776 | 0.490 | 0.959 | 0.918 | **5** |
| Los Angeles County | US-born citizens | 16210 | 0.143 | 0.857 | 0.184 | 0.490 | 0.510 | 0.408 | 0.184 | 41 |
| | Naturalised citizens | 5132 | 0.551 | 0.816 | 0.469 | 0.714 | 0.633 | 0.265 | 0.551 | 23 |
| | LPR | 1477 | 0.714 | 0.408 | 0.714 | 0.878 | 0.714 | 0.592 | 0.776 | 12 |
| | Documented temporary visa holders | 297 | 0.449 | 0.082 | 0.490 | 0.837 | 0.102 | 0.694 | 0.510 | 25 |
| | Undocumented immigrants | 886 | 0.918 | 0.429 | 0.857 | 0.980 | 0.776 | 0.939 | 0.980 | **2** |
| Orange County | US-born citizens | 4987 | 0.041 | 0.837 | 0.082 | 0.408 | 0.286 | 0.306 | 0.102 | 45 |
| | Naturalised citizens | 1498 | 0.224 | 0.633 | 0.388 | 0.612 | 0.245 | 0.061 | 0.286 | 36 |
| | LPR | 315 | 0.408 | 0.143 | 0.633 | 0.857 | 0.327 | 0.469 | 0.531 | 24 |
| | Documented temporary visa holders | 75 | 0.245 | 0.367 | 0.306 | 0.918 | 0.143 | 0.878 | 0.653 | 18 |
| | Undocumented immigrants | 126 | 0.857 | 0.449 | 0.918 | 1.000 | 0.306 | 1.000 | 1.000 | **1** |
| San Diego-Imperial | US-born citizens | 11559 | 0.082 | 0.735 | 0.061 | 0.265 | 0.408 | 0.163 | 0.122 | 44 |
| | Naturalised citizens | 2537 | 0.367 | 0.714 | 0.286 | 0.531 | 0.653 | 0.286 | 0.429 | 29 |
| | LPR | 869 | 0.673 | 0.571 | 0.694 | 0.816 | 0.469 | 0.551 | 0.694 | 16 |
| | Documented temporary visa holders | 107 | 0.102 | 0.224 | 0.367 | 0.898 | 0.061 | 0.653 | 0.327 | 34 |
| | Undocumented immigrants | 213 | 0.898 | 0.347 | 0.980 | 0.939 | 0.878 | 0.918 | 0.959 | **3** |

Values are vulnerability indices that range from 0 (least vulnerable) to 1 (most vulnerable) within each domain. Vulnerability indices scoring above the 75th percentile (0.75) are underlined. Sample is limited to adults aged 18 and over. Distinction between documented temporary visa holders and undocumented immigrants for years 2017–2020 is derived from multiple imputation using complete data in years 2015 and 2016.
Data source: Restricted data from the California Health Interview Survey (2015–2020).
*California's 2020 census regions. Source: https://census.ca.gov/regions/.
†Ranking is based on overall vulnerability. The 10 most vulnerable groups are bolded.

**Table 4** Concentration of relative social and COVID-19 vulnerability by immigrant status and California census region

| Region* | US-born citizens | Naturalised citizens | Legal permanent residents (LPR) | Documented temporary visa holders† | Undocumented immigrants† | Region average‡ |
|---|---|---|---|---|---|---|
| Superior California | 1 | 0 | 0 | 0 | 4 | 1.0 |
| North Coast | 2 | 2 | 1 | 1 | 4 | 2.0 |
| San Francisco Bay Area | 1 | 0 | 0 | 0 | 5 | 1.2 |
| Northern San Joaquin Valley | 2 | 0 | 2 | 0 | 3 | 1.4 |
| Central Coast | 1 | 0 | 1 | 0 | 4 | 1.2 |
| Southern San Joaquin Valley | 2 | 2 | 2 | 1 | 3 | 2.0 |
| Inland Empire | 2 | 2 | 1 | 1 | 4 | 2.0 |
| Los Angeles County | 1 | 1 | 1 | 1 | 5 | 1.8 |
| Orange County | 1 | 0 | 1 | 2 | 4 | 1.6 |
| San Diego-Imperial | 0 | 0 | 1 | 1 | 5 | 1.4 |
| Immigrant status group average‡ | 1.3 | 0.7 | 1.0 | 0.7 | 4.1 | 1.6 |

Values indicate the number of vulnerability themes scoring in the top 75th percentile across 50 nativity/immigration status-region groups. Higher numbers indicate higher relative vulnerability. The maximum possible value is 6. Sample is limited to adults aged 18 and over.
Data source: Restricted data from the California Health Interview Survey (2015–2020).
*California's 2020 census regions. Source: https://census.ca.gov/regions/.
†Distinction between documented temporary visa holders and undocumented immigrants for years 2017–2020 is derived from multiple imputation using complete data in years 2015 and 2016.
‡Unweighted average of the number of vulnerability themes scoring in the top 75th percentile.

access to healthcare go hand in hand and that these vulnerabilities are concentrated among undocumented immigrants living in Southern California. The heightened social vulnerabilities among undocumented immigrants are not unique to COVID-19. Researchers have used the same factors to determine vulnerabilities in a wide range of crises, including the 2004 tsunami in Aceh, Indonesia[55] and Hurricane Katrina in New Orleans.[56] Our analysis of sensitive immigration status data in the CHIS demonstrates how much undocumented immigrants are marginalised and disadvantaged, even in a state that arguably has the most inclusive policies towards immigrants.[29] Despite undocumented immigrants' greater social vulnerabilities, demographic and COVID-19-specific epidemiological risk factors were the highest among US citizens. These findings coincide with research that shows health advantages among recent immigrants that diminish to converge with US-born citizens over time.[57]

Our ecological approach also revealed regional disparities by immigration status. Such disparities may require parallel interventions to address the needs of a US-born population that is demographically and epidemiologically at risk for COVID-19, as well as an immigrant population that is healthy but socioeconomically disadvantaged.

Researchers and policy makers should interpret the findings with caution. First, the data collected were aggregated across the years leading up to the COVID-19 pandemic. Thus, the vulnerabilities may diverge from the pandemic's peak during the winter of 2020–2021. Still, the structural inequities that we measure in our analysis have been profound and persistent in immigrant communities.[30] Furthermore, increased immigration policy restrictions and heightened enforcement in the past 2 years have brightened the divisions between legal immigration statuses.[20] Second, the vulnerability indices are relative measures among California's 50 immigrant status-region

**Table 5** Correlation of vulnerability themes for nativity/immigration status-region groups

| | Socioeconomic vulnerability | Demographic vulnerability and disability | Minority status and language barriers | High housing density | Epidemiological risk factors | Low access to healthcare |
|---|---|---|---|---|---|---|
| Socioeconomic vulnerability | 1.000 | | | | | |
| Demographic vulnerability and disability | −0.421 | 1.000 | | | | |
| Minority status and language barriers | 0.858 | −0.603 | 1.000 | | | |
| High housing density | Insignificant at p<0.005 | −0.542 | 0.574 | 1.000 | | |
| Epidemiological risk factors | Insignificant at p<0.005 | 0.445 | Insignificant at p<0.005 | Insignificant at p<0.005 | 1.000 | |
| Low access to healthcare | 0.561 | −0.597 | 0.757 | 0.703 | −0.052 | 1.000 |

Correlations are calculated on vulnerability indices presented in table 2. Only values significant at the p<0.005 level are reported.
Data source: Restricted data from the California Health Interview Survey (2015–2020).

groups. Relative measures are more useful than absolute measures, however, when prioritising groups and regions.[46] The domain-specific measures also do not compare across domains. It does not identify whether socioeconomic vulnerability matters more for COVID-19 outcomes than, say, having low healthcare access. In the absence of prior knowledge on these domains' impact on COVID-19 outcomes, we have opted to place equal weight on each of the six domains. Third, the factors that we use in this study are not unilaterally associated with adverse outcomes from infection and disease progression. Some factors such as having an occupation in healthcare delivery can be both detrimental (ie, exposure to the virus) and protective (ie, income source and earlier access to vaccines). The factors are also not independent and can be connected in opposing directions. For example, people without a usual source of healthcare may be less likely to be diagnosed with comorbid conditions or work in healthcare occupations.

Despite these limitations, this article concretely examines immigrants' unique and diverse vulnerabilities associated with the COVID-19 pandemic. Population-representative analysis of undocumented immigrants by subregion is scarce, and our analysis aims to inform future disaster preparations.

## CONCLUSION

Exclusionary policies against immigrants have created a nation that stratifies its people based on legal immigration status.[58] Immigrants are weaved into society as family members, neighbours and coworkers of US-born citizens, and the consequences of ineffective public health measures among marginalised immigrants will spill over to everyone in the community.[35] In the absence of broad reform at the federal level, state and local governments must address the unique challenges immigrants face in their communities. Vaccination programmes must explicitly engage with immigrants who have tenuous ties with the healthcare system and are wary of interactions with the government. Safety-net programmes must be inclusive to all and actively overcome immigrants' reluctance to apply and enrol.

**Contributors** HS acquired funding and access to the restricted data, led the conceptualisation and the analysis and contributed to writing and editing of the manuscript. JKA contributed to the methodology, the literature review and the writing and editing of the manuscript. HS is responsible for the overall content as the guarantor.

**Funding** This project was supported by the Eunice Kennedy Shriver National Institute of Child Health and Human Development (NICHD, grant number R00HD096322).

**Competing interests** None declared.

**Patient and public involvement** Patients and/or the public were not involved in the design, or conduct, or reporting, or dissemination plans of this research.

**Patient consent for publication** Not required.

**Ethics approval** The use of data for this project has been approved by the UCLA South General Institutional Review Board (IRB number: 11-002227).

**Provenance and peer review** Not commissioned; externally peer reviewed.

**Data availability statement** Data may be obtained from a third party and are not publicly available. Restricted version of the California Health Interview Survey is managed and made available to researchers by UCLA Center for Health Policy Research, https://healthpolicy.ucla.edu/

**ORCID iDs**
Heeju Sohn http://orcid.org/0000-0001-5515-7011
Jasmine Ko Aqua http://orcid.org/0000-0002-4226-8748

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
