## [Reviewer comments · BMJ Open]

ARTICLE DETAILS

TITLE (PROVISIONAL)	Geographic Variation in COVID-19 Vulnerability by Legal Immigration Status in California: a pre-pandemic cross-sectional study
AUTHORS	Sohn, Heeju; Aqua, Jasmine

VERSION 1 – REVIEW

REVIEWER	Kathleen Page Johns Hopkins University, Medicine
REVIEW RETURNED	02-Aug-2021

GENERAL COMMENTS	The authors adapt the CDC’s social vulnerability index to add COVID-19 vulnerability variables and, using data from the California Health Interview Survey, assess COVID-19 vulnerability in California by immigration status and place of residence. The study’s main (and important) contribution is to highlight the heterogeneity in risk by immigration status and geography. There is a significant “data gap” in the health disparities literature about the vulnerabilities of undocumented migrants. Although there have been several qualitative studies and commentaries highlighting the disproportionate impact of COVID-19 in this population, there is very limited quantitative data on this topic, in part because of limitations in how data is collected and analysed. This study sheds some light on this issue, though the data analysed preceded the COVID-19 pandemic (data from 2014-2019) and is not linked or associated with COVID-19 outcomes (e.g. COVID positivity rates, morbidity or mortality). Strengths of the study include: • Large dataset with data variables that include immigration status and can stratify by undocumented, temporary visa holders, naturalized citizens and permanent residents, and underscores the particular vulnerabilities of undocumented migrants• Stratification by geographic location highlights the impact of geography on vulnerability index• Highlights interesting heterogeneity in the vulnerability domains by immigration status.o demographic vulnerability and disability are generally higher among US born individuals than undocumented immigrants, and epidemiologic vulnerabilities are also higher in some areas of the country among US born individuals than undocumented immigrants (there are some limitations that are described below)o other vulnerabilities (socioeconomic, minority/language, access to care) are higher for undocumented migrants• These data, and its granular analysis, can help inform health policy and interventions
---

	Weaknesses/suggestions for improvement include:  • While the study looks at a construct of COVID-19 vulnerability (adapted from the CDC social vulnerability index), the authors do not directly analyse COVID-19 data  o This may be for a future study, but census-track level data of COVID-19 rates could be used to assess the relationship between the COVID-19 Social Vulnerability Index domains by census track level and actual COVID-19 rates or mortality • COVID-19 vulnerability has come conflicting factors in the domains which may impact undocumented migrants in a very specific way, and could affect the calculation of vulnerability. For example:  o Epidemiologic risk factors (domain 5) includes chronic health conditions that predispose to severe COVID but these tend to be underdiagnosed in patients without access to care (Domain 6), which is more common among undocumented migrants ineligible for Medicaid or the ACA. o Socioeconomic vulnerability includes unemployment, but one of the most important risk factors for undocumented migrants early in the epidemic was not being unemployed- i.e. continuing to work in high risk jobs (construction, meat processing, etc) because they were ineligible for unemployment benefits or the stimulus check • Data limitations- undocumented status was only collected in 2015-2016 and imputed for the other years  o per authors only 2.29% of sample missing immigration status but would recommend specific statistical review to ensure methods are ok o Would also recommend adding n to the various categories in Table 2- we know that the dataset included 189,754 respondents, but have no idea how many fell in the various categories- this is important to understand context and analysis • Recommend updating introduction/background as follows  o Pubmed or other search using key terms “undocumented, COVID-19, US” or some similar permutation- while limited, there have been some publications on the impact of COVID-19 on undocumented migrants which would help set the stage for this analysis. There are no COVID-19 specific papers on this topic cited which is unusual. For example, the authors state “Undocumented immigrants face especially high risk for adverse COVID-19 outcomes” without a citing ref (of which there are several- I can provide if needed) o Clarify sections on access to COVID-19 care for undocumented migrants.  [ ] The authors state “The recent Coronavirus Aid, Relief, and Economic Security Act explicitly barred undocumented immigrants from receiving financial relief”. While it is true that undocumented immigrants could not receive cash assistance (stimulus checks), a large portion of the CARE Act included support for COVID-19 health care, and this include the care of undocumented immigrants (testing, hospital care, etc). Would recommend a better (primary citation) for this [ ] Would also clarify that vaccines are available to all, regardless of documentation status [ ] The authors talk about the public charge rule, which undoubtedly dampened healthcare utilization, but it should be acknowledged that the public charge rule did not apply to COVID-19 care (as early as March 2021, the USCIS issued a statement clarifying this- admittedly, many undocumented migrants were likely unaware of this change, but I do think that the introduction/background could focus a little more specifically on COVID-19 given the title of this paper) o Discussion
--	---

	□ The opening sentence states “our study quantifies the degrees to which undocumented migrants face disproportionate vulnerabilities during crises” but this is not entirely accurate as the data analysed preceded the COVID-19 pandemic and some factors may have changed during this period
--	---

REVIEWER	Susitha Wanigaratne SickKids Research Institute
REVIEW RETURNED	20-Sep-2021

GENERAL COMMENTS	This is an ecological study describing constructs related to social vulnerability to COVID-19, measured at the regional level, for population groups in California with a particular emphasis on undocumented immigrants. I appreciate the focus on undocumented immigrants and the emphasis on identifying where interventions to decrease social vulnerabilities may be most beneficial. I have a few suggestions, mainly with respect to clarity around methods and results as well as some minor comments. Suggestions  1. Please provide a descriptive table of population characteristics (e.g., age, sex etc - usually Table 1) by legal immigration status categories. Since this is a representative survey, please use weighted frequencies if possible. 2. Given the survey is based on random-digit dialing, can the authors provide some detail about if/how efforts were made to overcome likely barriers to participation? E.g., having access/being able to afford a phone, the likely reluctance of undocumented migrants to participate in the survey (e.g., given fear of deportation), language barriers etc. Given these possible barriers, can the authors also provide justification for the assessment of representativeness of the undocumented population? A comment in the limitations section may be necessary. 3. On page 8, line 52 – are “non-LPRs with valid US visas” referred to as “documented temporary visa holders” in other parts of the manuscript? Please be consistent in the labelling of groups for clarity. 4. What proportion of the groups labeled as “undocumented” + “documented temporary visa holders” had their immigration status imputed? Has this method been validated or have other studies used this method? I see several references which may support the approach – can the authors provide more details? Consider adding to limitations section of discussion as necessary. 5. Can the authors provide detail one which ecological variables are included in the CDC’s SVI index? Which of these variables could and could not be included based on the data available? Please provide justification for the variables that have been retained. To the authors point in the discussion about vulnerability to infection vs. severe disease, it would be helpful for readers if authors provided support/references for the relationship between each variable and either COVID-19 infection or severe outcomes in the methods section. 6. Given the referenced study supporting including additional ecological variables important to the pandemic is from India (ref #29), please provide justification for these additional variables for use in the US context given very different public health and policy contexts. 7. What is the rationale for combining occupational risks with a construct related to epidemiological risk factors? Biological risk factors (e.g., obesity) relate to severity and not SARS-CoV-2
---

	infection whereas it is the opposite for occupational risk factors. Furthermore, some immigrant groups may be less likely to have biological risk factors but have high occupational risk (i.e., people have to be physically healthy to do many jobs). Authors may want to consider creating a separate construct for occupational vulnerability. 8. Please describe the analytic approach and rationale for the information that populates table 3 and table 4 in the methods section of the manuscript, as it is not currently described. Based on the footnotes, I thought that the number of domains for which each immigration/region group scores >75th percentile (table 3) is summarized in table 4. But the information in table 4 is not consistent with this - looking at table 3, superior California – US born natives are vulnerable in 2 domains but table 4 indicates 1 domain. 9. What does the word “concentration” refer to in relation to the information presented in Table 4? 10. In the results section - the descriptions of table 3 and table 4 have been swapped (see page 14). E.g., table 3 describes summary measures, not R values whereas the results section for table 3 describes R values. 11. Given this study uses secondary data, was there an ethics review conducted by those who collected the data (e.g., UCLA etc)? If so, please include this information in the methods section. Minor comments  1. What does the word “sensitive” mean on page 7, line 8? 2. I recognize that different jurisdictions understand the word “native” differently – here I think it refers to people born in the United States regardless of Indigenous status. However, considering the US policies that have decimated and excluded or marginalized the surviving Indigenous peoples in California for hundreds of years, I wonder whether a different term for “native US population” could be considered here? US-born population instead?
--	--

VERSION 1 – AUTHOR RESPONSE

Reviewer: 1 Dr. Kathleen Page, Johns Hopkins University Comments to the Author:

The authors adapt the CDC’s social vulnerability index to add COVID-19 vulnerability variables and, using data from the California Health Interview Survey, assess COVID-19 vulnerability in California by immigration status and place of residence. The study’s main (and important) contribution is to highlight the heterogeneity in risk by immigration status and geography. There is a significant “data gap” in the health disparities literature about the vulnerabilities of undocumented migrants. Although there have been several qualitative studies and commentaries highlighting the disproportionate impact of COVID-19 in this population, there is very limited quantitative data on this topic, in part because of limitations in how data is collected and analysed. This study sheds some light on this issue, though the data analysed preceded the COVID-19 pandemic (data from 2014-2019) and is not linked or associated with COVID-19 outcomes (e.g. COVID positivity rates, morbidity or mortality).

Strengths of the study include:

- Large dataset with data variables that include immigration status and can stratify by undocumented, temporary visa holders, naturalized citizens and permanent residents, and underscores the particular vulnerabilities of undocumented migrants
- Stratification by geographic location highlights the impact of geography on vulnerability index

- **Highlights interesting heterogeneity in the vulnerability domains by immigration status.**
 - **demographic vulnerability and disability are generally higher among US born individuals than undocumented immigrants, and epidemiologic vulnerabilities are also higher in some areas of the country among US born individuals that undocumented immigrants (there are some limitations that are described below)**
 - **other vulnerabilities (socioeconomic, minority/language, access to care) are higher for undocumented migrants**
- **These data, and its granular analysis, can help inform health policy and interventions**

Weaknesses/suggestions for improvement include:

While the study looks at a construct of COVID-19 vulnerability (adapted from the CDC social vulnerability index), the authors do not directly analyse COVID-19 data

This may be for a future study, but census-track level data of COVID-19 rates could be used to assess the relationship between the COVID-19 Social Vulnerability Index domains by census track level and actual COVID-19 rates or mortality

Thank you for the comment. We are also looking forward to studying the impact of COVID-19 on immigrant populations in future studies!

COVID-19 vulnerability has come conflicting factors in the domains which may impact undocumented migrants in a very specific way, and could affect the calculation of vulnerability. For example:

Epidemiologic risk factors (domain 5) includes chronic health conditions that predispose to severe COVID but these tend to be underdiagnosed in patients without access to care (Domain 6), which is more common among undocumented migrants ineligible for Medicaid or the ACA.

Socioeconomic vulnerability includes unemployment, but one of the most important risk factors for undocumented migrants early in the epidemic was not being unemployed- i.e. continuing to work in high risk jobs (construction, meat processing, etc) because they were ineligible for unemployment benefits or the stimulus check

Thank you very much for this thoughtful feedback. We have expanded the Discussion section to acknowledge this limitation of the methodological approach.

Third, the factors that we use in this study are not unilaterally associated with adverse outcomes from infection and disease progression. Some factors such as having an occupation in health care delivery can be both detrimental (i.e., exposure to the virus) and protective (i.e., income source and earlier access to vaccines). The factors are also not independent and can be connected in opposing directions. For example, people without a usual source of health care may be less likely to be diagnosed with comorbid conditions or work in health care occupations. (Discussion)

Data limitations- undocumented status was only collected in 2015-2016 and imputed for the other years per authors only 2.29% of sample missing immigration status but would recommend specific statistical review to ensure methods are ok

Following Reviewer 2's suggestion, we have expanded the methods section to include a more detailed description of the imputation method. The approach has been studied and applied by immigration scholars in the past decade or so.

The imputation method to differentiate undocumented immigrants from documented temporary visa holders has been applied in national surveys such as the Survey of Income and Program Participation¹. It has also been applied to impute immigration status in a "recipient" survey (American Community Survey) using data from a "donor" survey that directly collected immigration information^{2,3}. These approaches are an extension of multiple imputation methods that leverage the relationships between variables with missing and known characteristics⁴. (Methods)

Would also recommend adding n to the various categories in Table 2- we know that the dataset included 189,754 respondents, but have no idea how many fell in the various categories- this is important to understand context and analysis

Thank you for the suggestion. We have added a column in Table 3 (revision of Table 2 as we added a new table to our manuscript) to show the unweighted number of observations in each immigrant status-region category.

Recommend updating introduction/background as follows

Thank you very much for the very helpful feedback. We have extensively updated our background and literature section to reflect the dynamic COVID-19 pandemic as well as immigration policy. Much has changed since we submitted the manuscript in early 2021. We will not copy-paste the entire section again here, but we highlight key sections most relevant to comments below.

Pubmed or other search using key terms "undocumented, COVID-19, US" or some similar permutation- while limited, there have been some publications on the impact of COVID-19 on undocumented migrants which would help set the stage for this analysis. There are no COVID-19 specific papers on this topic cited which is unusual. For example, the authors state "Undocumented immigrants face especially high risk for adverse COVID-19 outcomes" without a citing ref (of which there are several- I can provide if needed)

As the COVID-19 pandemic progressed, commentaries and limited research described the unique difficulties and high risk for adverse COVID-19 outcomes that undocumented immigrants faced. Reports from hospital emergency rooms detailed inpatient teams struggling to communicate with Spanish-speaking patients using language lines and through layers of personal protective equipment⁵. In addition to reports of language barriers, a cross-sectional survey of adult, Spanish-speaking, non-citizen Latinx immigrants found that a substantial percentage of participants would not identify an undocumented household member or coworker during contact tracing, believed that uninsured immigrants were limited to hospital emergency departments for COVID-19 testing or treatment, and agreed that using public COVID-19 testing and treatment services could jeopardize an individual's immigration prospects⁶. Reports also noted high COVID-19 case rates and numerous significant outbreaks in federal immigrant detention centers⁷⁻⁹, as well as a fear that convention centers that served as COVID-19 treatment facilities were actually immigration detention centers⁵.

Clarify sections on access to COVID-19 care for undocumented migrants.

The authors state “The recent Coronavirus Aid, Relief, and Economic Security Act explicitly barred undocumented immigrants from receiving financial relief”. While it is true that undocumented immigrants could not receive cash assistance (stimulus checks), a large portion of the CARE Act included support for COVID-19 health care, and this include the care of undocumented immigrants (testing, hospital care, etc). Would recommend a better (primary citation) for this

California became the first state to provide COVID-19 disaster relief assistance to undocumented adults who are ineligible for other forms of assistance, providing a one-time direct assistance payment of \$500 (maximum of \$1000 per household) ¹⁰. This \$75 million dollar disaster relief assistance program was estimated to reach 150,000 undocumented Californians through twelve immigrant-serving nonprofit organizations ¹⁰. The demand for relief quickly overwhelmed the available resources, with reports of people unable to get through phone lines due to extremely high call volumes and equating the direct assistance to “winning the lottery” ¹¹.

Would also clarify that vaccines are available to all, regardless of documentation status

The authors talk about the public charge rule, which undoubtedly dampened healthcare utilization, but it should be acknowledged that the public charge rule did not apply to COVID-19 care (as early as March 2021, the USCIS issued a statement clarifying this- admittedly, many undocumented migrants were likely unaware of this change, but I do think that the introduction/background could focus a little more specifically on COVID-19 given the title of this paper)

Though USCIS issued an alert on March 14, 2020, that COVID-19 testing, treatment, preventive care, and vaccines (when available) would be exempt from the “public charge” admissibility assessment and stopped applying this Public Charge Final Rule on March 9, 2021 ¹², communication and implementation of these changes remained unclear, and many immigrants still believed that using public COVID-19 testing and treatment services could jeopardize their immigration prospects ⁶. When COVID-19 vaccines became available to the general public, the Department of Homeland Security (DHS) released a statement that they fully support equal access to the vaccine for undocumented immigrants ¹³. Despite this public statement, the majority of state public health websites do not explicitly mention this information, and many reports emerged throughout 2021 of undocumented immigrants being asked to provide Social Security Numbers at vaccination sites or being turned away for not presenting a state-issued ID ¹⁴⁻¹⁶. Many undocumented immigrants also occupy jobs without paid leave and have language barriers that can impact their comprehension of vaccine information and education, introducing additional structural barriers to vaccination.

Discussion

The opening sentence states “our study quantifies the degrees to which undocumented migrants face disproportionate vulnerabilities during crises” but this is not entirely accurate as the data analysed preceded the COVID-19 pandemic and some factors may have changed during this period

Thank you for the suggestion. We have revised the opening sentence to as follows.

Our study highlights the unequal social vulnerabilities between people with different legal immigration statuses across California during the years leading up to the COVID-19 pandemic.

Reviewer: 2 Dr. Susitha Wanigaratne, SickKids Research Institute Comments to the Author:

This is an ecological study describing constructs related to social vulnerability to COVID-19, measured at the regional level, for population groups in California with a particular emphasis on undocumented immigrants. I appreciate the focus on undocumented immigrants and the emphasis on identifying where interventions to decrease social vulnerabilities may be most beneficial. I have a few suggestions, mainly with respect to clarity around methods and results as well as some minor comments.

Suggestions

Please provide a descriptive table of population characteristics (e.g., age, sex etc - usually Table 1) by legal immigration status categories. Since this is a representative survey, please use weighted frequencies if possible.

Thank you very much for the suggestion. We have added a new table (Table 2) with descriptive characteristics (weighted) by immigrant status. In this new table, we also noted which factors we used to determine vulnerabilities in our analysis.

Given the survey is based on random-digit dialing, can the authors provide some detail about if/how efforts were made to overcome likely barriers to participation? E.g., having access/being able to afford a phone, the likely reluctance of undocumented migrants to participate in the survey (e.g., given fear of deportation), language barriers etc. Given these possible barriers, can the authors also provide justification for the assessment of representativeness of the undocumented population? A comment in the limitations section may be necessary.

A unique strength of the CHIS over many national-level data lies in its ability to produce estimates for smaller racial/ethnic and immigrant groups. To achieve this objective, the CHIS employed an address-based sample design (44 geographic strata nested in the State's 58 counties) and over-sampled from places with high likelihoods of having underrepresented residents. In recent years, CHIS specifically targeted households with Asians (with a particular focus on Korean and Vietnamese), Hispanic or Spanish-speaking persons, and non-US citizens. The CHIS also conducted the survey in Spanish, Vietnamese, Korean, Mandarin, Cantonese, and Tagalog in addition to English to not only increase the response rates among non-English speakers but also to improve interview quality. These languages were identified as the most spoken languages by non-English speakers in California from the 2010 US Census. In the 2019 CHIS, about 72 percent of non-English responses were conducted in Spanish. The Chinese dialects were the second most frequently used languages at 16 percent. About 5 percent of adult respondents in the 2019 CHIS opted for the English version of the survey.

Prior surveys have successfully applied the "peel the onion" method for discerning respondents' immigration information that the CHIS used in 2015 and 2016. One notable example is the Los Angeles Families and Neighborhood Survey (LA FANS), which used the same set of questions as the CHIS to determine whether a respondent was undocumented. An auxiliary study that specifically examined the responses to immigration-related questions found that Spanish-speaking respondents who were interviewed by Spanish-speaking interviewers reported little unease in replying to immigration-related questions¹⁷. Furthermore, despite being given the option to skip or not respond, the item-specific response rates were high¹. The increase in anti-immigrant rhetoric and exclusionary policies in 2017 may have created more hesitation among undocumented immigrants in recent surveys. The CHIS, however, did not ask for specific visa information beyond naturalization and LPR status during those years, and the imputation models may have circumvented potential sources of response bias.

We have included a condensed description of CHIS’s efforts to improve response rates among underrepresented immigrants living in California.

On page 8, line 52 – are “non-LPRs with valid US visas” referred to as “documented temporary visa holders” in other parts of the manuscript? Please be consistent in the labelling of groups for clarity.

Thank you for the comment. We have standardized the language to “documented temporary visa holders” throughout the manuscript to describe the group of people who were not LPRs nor undocumented immigrants identified from the CHIS.

What proportion of the groups labeled as “undocumented” + “documented temporary visa holders” had their immigration status imputed? Has this method been validated or have other studies used this method? I see several references which may support the approach – can the authors provide more details? Consider adding to limitations section of discussion as necessary.

Thank you very much for the suggestion. We have added more details to the imputation methodology in our revised manuscript.

*The imputation method to differentiate undocumented immigrants from documented temporary visa holders has been applied in national surveys such as the Survey of Income and Program Participation¹. It has also been applied to impute immigration status in a “recipient” survey (American Community Survey) using data from a “donor” survey that directly collected immigration information^{2,3}. These approaches are an extension of multiple imputation methods that leverage the relationships between variables with missing and known characteristics⁴. (**Methods**)*

We have also added more precise numbers on the number of observations whose immigration status was imputed in the manuscript.

	Not imputed	Imputed	Total
US-born citizens	100,387	0	100,387
Naturalized citizens	18,386	0	18,386
Legal permanent residents (LPR)	5,825	0	5,825
Documented temporary visa holders	1,455	1,358	2,813
Undocumented immigrants	374	743	1,117
Total	126,427	2,101	128,528
(% of total)	98.4%	1.6%	

Can the authors provide detail on which ecological variables are included in the CDC’s SVI index? Which of these variables could and could not be included based on the data available?

Please provide justification for the variables that have been retained. To the authors point in the discussion about vulnerability to infection vs. severe disease, it would be helpful for readers if authors provided support/references for the relationship between each variable and either COVID-19 infection or severe outcomes in the methods section.

Thank you for the suggestion. We have revised the data section to describe how we adapted the CDC's Social Vulnerability Index (SVI). We were able to include all of CDC's factors except two: disability due to physical or mental disability and vehicle ownership. Instead, we added factors that indicated the level of severe psychological distress (based on Kessler's questionnaire) and the degree of urbanization.

We adapted the validated U.S. Centers for Disease Control and Prevention (CDC) 's Social Vulnerability Index (SVI)⁴⁰ to develop a COVID-19 vulnerability index. We modified the CDC SVI based on the variables available in our dataset and expanded the index to include additional factors critical to the COVID-19 pandemic ⁴¹. The first four themes in our COVID-vulnerability index (socioeconomic, demographic/disability, minority and language, and housing density) are based on CDC's SVI. We were not able to include two factors from CDC's SVI—physical/mental/emotional disability status and vehicle ownership—as CHIS did not ask these questions during the study period. Instead, we included a factor that indicated serious psychological distress based on Kessler's Psychological Distress Questionnaire⁴². We also added a factor variable indicating the proportion of respondents living in an urban area to augment the CDC's housing density theme. In addition to the CDC's SVI four original themes, our analysis utilizes CHIS's detailed health questionnaire and examined two more themes: epidemiological factors and access to health care. Overall, we incorporated 21 factors across six domains in our COVID-19 vulnerability index. Table 1 lists the six themes and their factors. (Methods)

Given the referenced study supporting including additional ecological variables important to the pandemic is from India (ref #29), please provide justification for these additional variables for use in the US context given very different public health and policy contexts. What is the rationale for combining occupational risks with a construct related to epidemiological risk factors? Biological risk factors (e.g., obesity) relate to severity and not SARS-CoV-2 infection whereas it is the opposite for occupational risk factors. Furthermore, some immigrant groups may be less likely to have biological risk factors but have high occupational risk (i.e., people have to be physically healthy to do many jobs). Authors may want to consider creating a separate construct for occupational vulnerability.

We appreciate the comment and suggestion. We did not include variables from Acharya et al.'s India study that does not apply to the US context. The only variable that we directly adapted was the proportion of respondents living in an urban area to augment the Housing Density domain. Instead of using whether communities had health care clinics that Acharya and colleagues used, we used proportions uninsured or not having a usual source of care as factors in the Access to Health Care domain.

We made the decision to combine occupational risk with biological risk into one category to reflect the vulnerability that is more specific to COVID-19 than the traditional socioeconomic determinants of health in the CDC's index. As both reviewers note, the individual factors within this category (as well as across categories) can reasonably influence the risk of COVID-19 infection and complication in opposing directions. For example, having a job in healthcare can increase the likelihood of contracting COVID-19 but also lowers socioeconomic vulnerability due to unemployment or low income. Furthermore, the same factors' impact of COVID-19 risk can change over time. After the advent of COVID-19 vaccines, those who worked in high-risk occupations (e.g., healthcare) or had comorbid

conditions (e.g., cardiovascular conditions or diabetes) gained access earlier than others. We acknowledge this limitation to the approach in studying vulnerability and have added a section in our revised Discussion.

Please describe the analytic approach and rationale for the information that populates table 3 and table 4 in the methods section of the manuscript, as it is not currently described. Based on the footnotes, I thought that the number of domains for which each immigration/region group scores >75th percentile (table 3) is summarized in table 4. But the information in table 4 is not consistent with this - looking at table 3, superior California – US born natives are vulnerable in 2 domains but table 4 indicates 1 domain.

The description in the footnotes is correct. In our revision, we have checked that the number of immigration/region groups with scores > 75th percentile in (new) Table 3 is accurately reflected in (new) Table 4.

What does the word “concentration” refer to in relation to the information presented in Table 4? In the results section - the descriptions of table 3 and table 4 have been swapped (see page 14). E.g., table 3 describes summary measures, not R values whereas the results section for table 3 describes R values.

We have revised the description of tables 3 and 4 in the results section to better reflect the ordering of the tables. We have also added language to clarify “concentration of vulnerability” in table 3.

Table 4 presents the concentration of vulnerability for each immigrant status-region group. The values in Table 4 indicate the number of vulnerability themes out of a possible six that scored in the top 75th percentile. Table 5 presents a full correlation table between the six themes with tests of statistical significance. Groups whose vulnerability stems from low socioeconomic status are likely to share vulnerabilities from being a member of a minority group, experiencing language barriers ($R=0.858$), and having low access to health care ($R=0.561$). Groups' minority populations and language barriers are also correlated with high housing density ($R=0.574$) and low access to health care ($R=0.757$). High epidemiological and demographic vulnerabilities were not significantly correlated with high vulnerabilities from social causes. (Results)

Given this study uses secondary data, was there an ethics review conducted by those who collected the data (e.g., UCLA etc)? If so, please include this information in the methods section.

We have added the IRB board and number that approved the use of CHIS data for this study in the data section as well as in a separate ethics statement.

The use of the data for this project was approved by UCLA's South General IRB (IRB #11-002227).

Minor comments

What does the word “sensitive” mean on page 7, line 8?

We have revised the paragraph for clarity:

The CHIS is a large, annual random-digit telephone survey of public health and health care access issues in California and is one of few representative surveys of this scale that collected information on detailed immigrant documentation status uncommon in large-scale surveys. (Data)

I recognize that different jurisdictions understand the word “native” differently – here I think it refers to people born in the United States regardless of Indigenous status. However, considering the US policies that have decimated and excluded or marginalized the surviving Indigenous peoples in California for hundreds of years, I wonder whether a different term for “native US population” could be considered here? US-born population instead?

Thank you very much for the thoughtful suggestion. We have revised our language throughout the manuscript and use the terms “US-born citizens” or “US-born population”.

VERSION 2 – REVIEW

REVIEWER	Kathleen Page Johns Hopkins University, Medicine
REVIEW RETURNED	19-Jan-2022
GENERAL COMMENTS	The authors have addressed the reviewers' concerns adequately and in my view is acceptable for publication.